# When a Rapid Accurate Diagnosis Changes Therapeutic Approach: Recognizing Acute Abdominal Pain with Ascites as a Possible Presentation of Systemic Lupus Erythematosus

**DOI:** 10.3390/diagnostics12112605

**Published:** 2022-10-27

**Authors:** Szu-Cheng Huang, Yi-Ling Chan, Hao-Tsai Cheng, Zhong Ning Leonard Goh, Yon-Cheong Wong, Chen-Ken Seak, Joanna Chen-Yeen Seak, Chih-Huang Li, Hsien-Yi Chen, Chen-June Seak

**Affiliations:** 1Department of Emergency Medicine, Lin-Kou Medical Center, Chang Gung Memorial Hospital, Taoyuan 33305, Taiwan; 2College of Medicine, Chang Gung University, Taoyuan 33302, Taiwan; 3Department of Gastroenterology and Hepatology, Lin-Kou Medical Center, Chang Gung Memorial Hospital, Taoyuan 33305, Taiwan; 4Department of Gastroenterology and Hepatology, New Taipei Municipal Tucheng Hospital, New Taipei City 23652, Taiwan; 5Sarawak General Hospital, Kuching 93586, Sarawak, Malaysia; 6Department of Medical Imaging and Intervention, Lin-Kou Medical Center, Chang Gung Memorial Hospital, Taoyuan 33305, Taiwan; 7Department of Emergency Medicine, New Taipei Municipal Tucheng Hospital, New Taipei City 23652, Taiwan

**Keywords:** Systemic lupus erythematosus, lupus mesenteric vasculitis, comb sign, emergency department, serum-ascites albumin gradient, anti-double stranded DNA

## Abstract

Systemic lupus erythematosus (SLE) is a chronic, multi-organ autoimmune disease which rarely presents with peritoneal involvement. As such, its diagnosis in the emergency department (ED) based on a clinical presentation of gastrointestinal symptoms is extremely challenging. Yet, reaching such a diagnosis in the ED is crucial for avoiding unnecessary surgical intervention and initiating early glucocorticoid therapy to maximise patient outcomes. Here, we report a case of newly diagnosed SLE in a 28-year-old lady who presented atypically and unusually with abdominal pain and ascites. She required extensive but methodical investigations, and was eventually diagnosed with lupus mesenteric vasculitis with underlying newly diagnosed SLE in the ED. The patient was promptly treated with methylprednisolone resulting in marked clinical improvement. Emergency physicians should be mindful of abdominal pain with ascites as an extremely rare but important clinical presentation of SLE. Early diagnosis and commencement of glucocorticoid therapy in these patients are crucial in halting disease progression and averting the need for surgical intervention.

## 1. Introduction

Systemic lupus erythematosus (SLE) is a chronic, multi-organ autoimmune disease. It frequently involves the gastrointestinal system and serosal tissues [1], while peritoneal involvement is very rarely described [2]. The symptoms of SLE are usually vague and non-specific, contributing to the challenge of diagnosing it in the emergency department (ED). Nevertheless, its early diagnosis in the ED is important in early initiation of glucocorticoid therapy to optimise patient outcomes. Here, we report a case of newly diagnosed SLE who presented atypically to our ED with vague gastrointestinal symptoms and was successfully treated with steroids, avoiding the need for unnecessary surgical intervention.

## 2. Case Presentation

Our patient, a 28-year-old lady with no prior medical illnesses, presented to our ED with a one-day history of persistent, diffuse, dull abdominal pain associated with nausea, vomiting, and watery stools. Physical examination revealed that she was tachycardic (pulse rate 115 beats/min) and slightly hypertensive (systolic blood pressure 143 mmHg), but otherwise not tachypneic nor febrile. There was generalised abdominal tenderness without any rebound tenderness. Laboratory investigations demonstrated leukocytosis with neutrophilia (white cell count 16 × 10^9^/µL, neutrophils 86.2%) and elevated levels of C-reactive protein (21.1 mg/dL). Other blood parameters were normal. Chest X-ray found mild left pleural effusion (Figure 1). The patient was initially treated for the provisional diagnosis of infectious gastroenteritis and colitis with empirical antibiotic therapy, but her symptoms of abdominal distension and pain continued to worsen.

Further blood investigations revealed hypoalbuminaemia (2.86 g/dL), with normal coagulation profile, amylase, lipase, and B-type natriuretic peptide. An abdominal computed tomography (CT) scan was performed which demonstrated extensive small bowel wall thickening with enhancement (target sign), and engorgement of visible mesenteric vessels (comb sign) with significant ascites; there were no ischaemic changes nor thrombosis of the mesenteric vessels (Figure 2). No free air suggestive of viscus perforation was seen. Paracentesis was subsequently performed to obtain ascitic fluid for analysis, with the following results: total protein 5.2 g/dL, albumin 2.36 g/dL, and negative bacteriology studies (Gram stain, acid-fast bacilli smear, Mycobacterium tuberculosis culture). Serum-ascites albumin gradient (SAAG) was 0.5 g/dL, prompting further investigation into peritoneal causes of the patient’s ascites.

Since prior investigations were not suggestive of infection or malignancy, tests to look for autoimmune causes were ordered. Anti-double stranded DNA (anti-dsDNA) antibodies via enzyme-linked immunosorbent assay, anti-Ro antibodies, anti-nuclear antibodies, anti-ribonucleic protein antibodies, lupus anticoagulant, and direct Coombs’ tests returned positive, with decreased C3 and C4 complement levels. Further targeted physical examination revealed polyarthritis over bilateral metacarpophalangeal and proximal interphalangeal joints, left knee, and bilateral ankle joints. As such, our patient fulfilled 7 out of the 17 Systemic Lupus International Collaborating Clinics (SLICC) 2012 criteria, with a total 2019 European League Against Rheumatism/American College of Rheumatology (EULAR/ACR) score of 27. A diagnosis of lupus mesenteric vasculitis (LMV) with newly diagnosed underlying SLE was made in the ED. She was treated promptly with intravenous methylprednisolone 40 mg 12-hourly for three days with subsequent tapering to oral prednisolone 20 mg twice daily upon discharge, for a total steroid therapy duration of three months. Oral hydroxychloroquine 200 mg once daily was also prescribed. Her abdominal pain was substantially alleviated, and she was discharged uneventfully after 10 days of hospitalisation.

Our patient was followed up in the outpatient clinic a month post-discharge with complete resolution of gastrointestinal symptoms and minimal arthralgia, though at two months post-discharge she developed another flare of polyarthritis and rashes which required a second course of steroids with coadministration of methotrexate. Repeat blood investigations at one month post-discharge demonstrated a decrease in anti-dsDNA antibody levels and increase in C3 and C4 complement levels. The results of these laboratory investigations have been summarized in Table 1.

## 3. Discussion

Coming to the diagnosis of SLE is particularly challenging owing to its heterogeneity of clinical presentations. Yet, early diagnosis and prompt treatment is crucial in preventing undesirable prognoses [3], especially in the ED where abdominal pain is a very common presenting symptom [4]. Ascites can be the presenting complaint in 8% to 11% of adult SLE patients. Therefore, SLE should be considered after evaluations of the cardiovascular, hepatic, and renal functions return normal. Ascitic fluid analysis with SAAG assessment might prove useful in further deductions into the root cause of ascites.

Given that SLE is such a challenging disease to diagnose, several criteria have been developed in an attempt to better identify and characterise it. The two latest criteria in use currently are the 2012 SLICC criteria with 96.7% sensitivity and 83.7% specificity, and the 2019 EULAR/ACR criteria with 96.1% sensitivity and 93.4% specificity [5,6]. Our patient fulfilled the minimum required scores for both criteria to be diagnosed with SLE—7 out of 17 SLICC criteria (joint disease, pleural effusion, antinuclear antibody, anti-dsDNA, lupus anticoagulant, low complement, and direct Coomb’s test) and 27 points on the EULAR/ACR criteria (autoimmune hemolysis, pleural effusion, joint involvement, lupus anticoagulant, low C3 and low C4, and anti-dsDNA).

C-reactive protein was elevated at 21.1 mg/dL in our patient, with leukocytosis of 16 × 10^9^/µL. While these results might not be typical of active SLE and suggest a possible bacterial infection, they can also be indicative of ongoing inflammation. These findings may have been due to concomitant acute gastroenteritis in our patient for which empirical antibiotics were given.

Comb sign was found on CT imaging of our patient’s abdomen. This sign refers to the hypervascular mesenteric appearance, usually seen in acute inflammatory conditions of the bowel and mesentery. In hindsight, the presence of comb sign in our patient’s abdominal CT scans is indicative of LMV [7]. Imaging, in particular CT scans, is the main modality for diagnosing LMV [8,9].

LMV is a severe and potentially life-threatening complication of SLE that requires urgent treatment, as mesenteric thrombosis and infarction can lead to bowel perforation and peritonitis [1,9,10]. Our patient fortunately presented before the development of such sequelae, and was successfully treated with high-dose glucocorticoids to achieve satisfactory disease control. SLE patients with gastrointestinal involvement often respond well to such therapy with good prognoses because of its inflammatory aetiology, with immune-complex deposition implicated in its pathogenesis [9]. Failure of steroid therapy that results in disease progression and/or bowel infarction with subsequent perforation may require surgical intervention [1,9].

An important point to note is that SLE is often a diagnosis of exclusion. Hence, SAAG is valuable in the initial evaluation of patients presenting with ascites. A high gradient (SAAG > 1.1 g/dL) is suggestive of portal hypertension, while a low gradient (SAAG < 1.1 g/dL) is indicative of a peritoneal cause. Since our patient had a SAAG of 0.5 g/dL, it prompted us to investigate further with regard to autoimmune diseases, with the eventual diagnosis of LMV.

It must also be noted that there are risks to inappropriate steroid administration in patients with acute abdomen of unclear aetiology. Such risks include worsening of occult sepsis and progression of ongoing infection, as well as the gastrointestinal side effects of peptic ulcers and upper gastrointestinal bleeding. Infections should be ruled out prior to steroid initiation, while clinicians may choose to employ proton pump inhibitors or histamine H_2_ blockers for gastroprotection to reduce the risk of peptic ulcers; our patient was placed on famotidine for the duration of her steroid therapy.

Laboratory investigations for autoantibodies crucial to the evaluation of patients for autoimmune diseases typically have a longer turnaround time of a few days. This may pose a diagnostic dilemma while waiting for results to return. We were nevertheless able to obtain our patient’s C3/C4 results within a day, right from the ED. In view of the decreased C3/C4 levels, together with a lack of evidence of infection and response to antibiotics, we investigated further toward autoimmune aetiologies of our patient’s clinical presentation. The detection of ANA and anti-DSDNA antibodies subsequently provided additional support for our clinical diagnosis of SLE.

Emergency physicians should consider the differential diagnoses of LMV and other gastrointestinal manifestations of SLE in young female patients presenting with new-onset ascites, especially after other possible causes have been ruled out. Early diagnosis and commencement of glucocorticoid therapy in the ED for these patients are crucial in halting disease progression and avoiding the need for unnecessary surgical intervention.

## Figures and Tables

**Figure 1 diagnostics-12-02605-f001:**
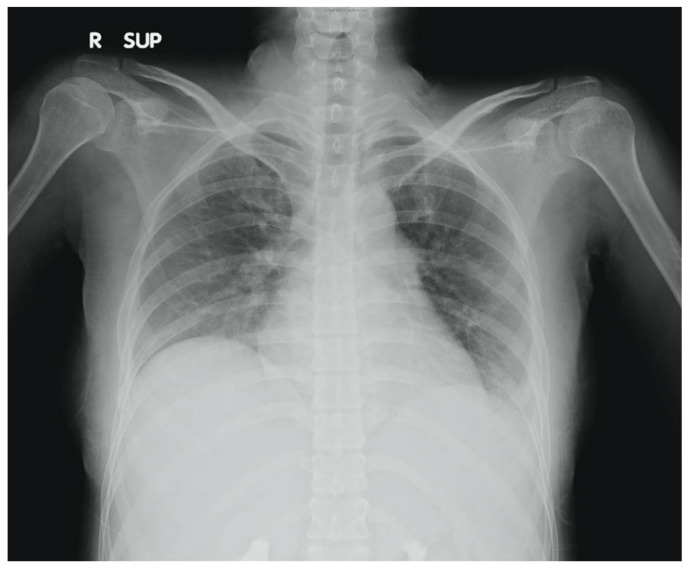
Chest X-ray of patient showing mild left pleural effusion.

**Figure 2 diagnostics-12-02605-f002:**
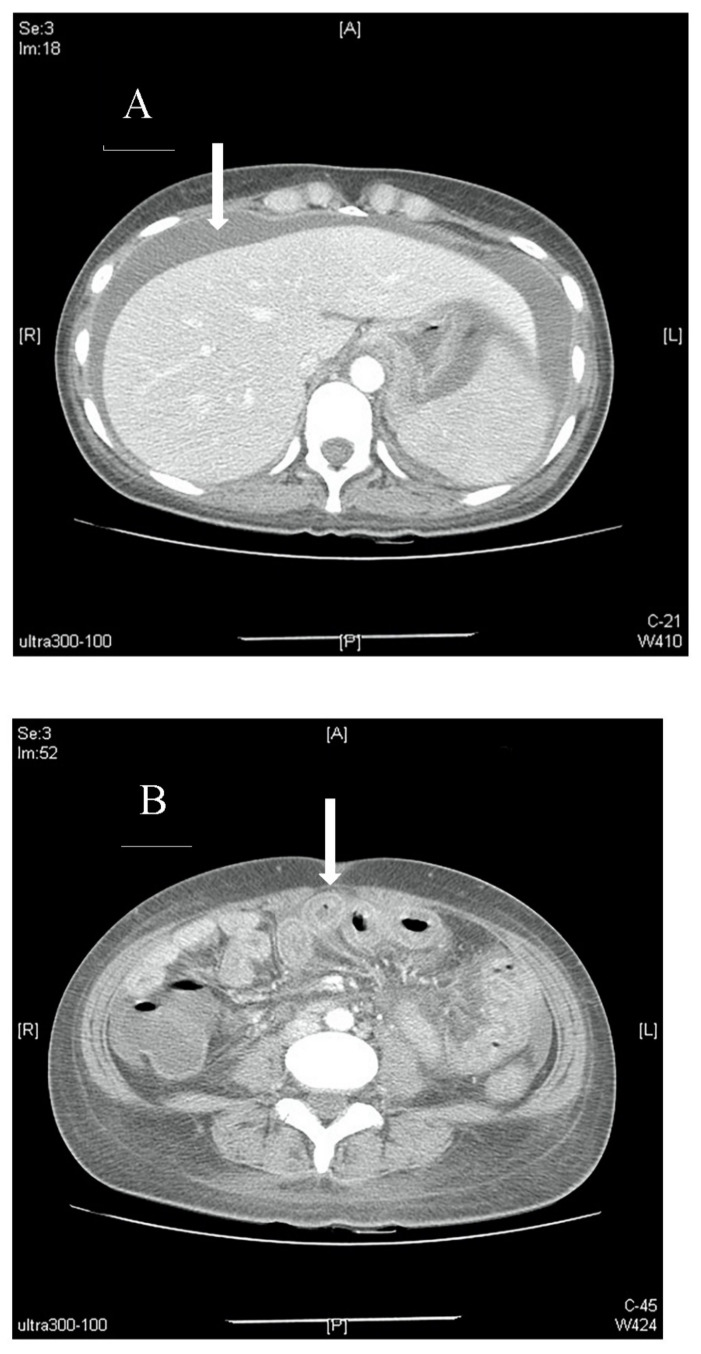
(**A**). Axial view of a contrast-enhanced abdominal CT scan showing massive ascites (arrow) with no signs of occlusion or filling defect in major vessels. (**B**). Axial view of a contrast-enhanced abdominal CT scan showing bowel-wall thickening and enhancement, which is also known as target sign (arrow). (**C**). Axial view of a contrast-enhanced abdominal CT scan showing edematous ileum wall with engorgement of mesenteric vessels and increased number of visible vessels, which is also known as comb sign (arrow).

**Table 1 diagnostics-12-02605-t001:** Summary of key laboratory investigation results during initial admission and on 1-month follow-up.

Laboratory Investigation	Reference Range	Initial Admission	On 1-Month Follow-Up
Anti-dsDNA antibodies (units/mL)	<92.6	492	273
Complement C3 (mg/dL)	90–180	27	62.1
Complement C4 (mg/dL)	10–40	4.77	9.63
Anti-nuclear antibodies	≤1:80	>1:1280, speckled (AC-4,5)	-
Anti-Smith antibodies (units/mL)	<5	10	-

## Data Availability

All data are available within the article.

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
