# Peer review of "When a Rapid Accurate Diagnosis Changes Therapeutic Approach: Recognizing Acute Abdominal Pain with Ascites as a Possible Presentation of Systemic Lupus Erythematosus"

_diagnostics, 2022, doi:10.3390/diagnostics12112605_

Round 1
Reviewer 1 Report
The authors presented a case of newly diagnosed systemic lupus erythematosus in a 28-year-old lady who presented atypically with abdominal pain and ascites.
Report is poorly written. Intdoduction is short and not focusing on a main objective. Case presentation is also poorly written. Figures are not important. Discussion is general, mostly repeating well-known facts from the literature. English needs improovement.
Although it is rare this condition is well-known and does not represent any novelty. I do not see any benefits for the readers of the Diagnostics from this report.
Author Response
We regret that the Reviewer is of the opinion that our manuscript holds no benefit for readers of Diagnostics. We beg to differ, as we think that it illustrates an important lesson that unnecessary surgical procedures can be avoided in a certain presentation of acute abdominal pain with ascites. We have made several amendments as per the suggestions of other reviewers and we hope that Reviewer #1 might change their opinion of our manuscript.
Reviewer 2 Report
very good and systematic manuscropt - nothing to add, accepted from my side!
Author Response
Thank you for the kind comments and positive review.
Reviewer 3 Report
Dear Authors,
You describe the case of a 28-year-old female patient who presented to your emergency department with abdominal discomfort and ascites.
With concomitant joint symptoms, low complement, positive Coombs test, and evidence of DNA antibodies, anti-Ro antibodies, anti-RNP, and lupus anticoagulant, you made the diagnosis of SLE and gave steroids with improvement of symptoms.
Increased visibility of the mesenteric vessels on CT you evaluate as lupus mesenteric vasculitis.
Acute abdomen is indisputably challenging and with evidence of fluid and high CRP, surgical intervention with laparotomy/laparoscopy is suggested. Cortisone prevented this in this case.
For a good case presentation, I would like to see the most important laboratory findings as a value in addition to the CT images, preferably in tabular form, and in the case of ANA also the fluorescence pattern. In addition, a progression of the most important parameters would be useful to underline the response.
The most reliable finding that "proves" ascites as associated with SLE is the LE cell phenomenon (phagocytized erythrocytes). Has this been tested or demonstrated?
Leukocytosis and markedly elevated CRP are atypical of active SLE as a cause and are suggestive of a bacterial infection. How do you explain these values?
Unfortunately, the case suggests that in "unclear" abdomen cortisone is preferable to surgery, which may be true in individual cases, but in most cases (an appendicitis, diverticulitis, enteritis or peritonitis is much more common) can lead to a significant worsening of prognosis.
A good paper should point out the risks of incorrect steroid therapy in unclear abdomen.
The steroid dose and regimen of steroid therapy should be named.
Detection of autoantibodies is usually not an emergency diagnostic test and is rarely obtained faster than in 2-3 working days. Without it, the diagnosis of SLE would not be tenable. How could this be achieved in your emergency department? What should be done if the values are not available?
ANA and also DNA antibodies can be false positive especially in infections. this should be considered.
The lupus classification criteria cited are not diagnostic criteria. The diagnosis is a medical decision.
If the patient has newly diagnosed SLE, as reported, therapy should include not only steroids but also hydroxychloroquine and vitamin D. If the patient also has arthritis, pleurisy, autoimmune hemolysis, and a lupus anticoagulant, this is not sufficient and immunosuppression is required. No position is taken on this. However, long-term planning should begin with the diagnosis to avoid new complications.
Lupus anticoagulant is described as positive, you should explicitly refer to the exclusion of thrombosis.
The diagnosis of vasculitis requires the detection of inflammatory cells in the vessel wall. The CT image can at most give an indication of a possible LMV.
In summary, I recommend a thorough revision of the case report so that readers can better weigh the benefits and risks of calculated steroid therapy for unclear abdomen in future similar decisions.
Reviewer 4 Report
I read with interest the article by Szu-Cheng Huang et al,who described a newly diagnosed lupus patient initial presented with acute abdominal pain with ascites. Eventually the patient got prompt diagnosis and GC treatment. I totally agree with the author's opinion that we need to pay attention to the GI involvement in lupus patients. In our clinical practice, one of clinical menifestations of the lupus enteritis is associated with hypercoagulation. So anti-coagulation therapy is very important for this subset patients, especially for lupus mesenteric vasculitis. I will require the author to add information regarding D-dimer level, haemostatic function, and anti-coagulation therapy.
Author Response
Thank you for the pertinent points raised in your review. We did not check the patient’s D-dimer levels, nevertheless on CT imaging the bowel walls were well enhanced with fair perfusion and no evident ischaemic changes. There was also no superior mesenteric arterial nor venous thrombosis seen on CT. The patient was not started on any anticoagulation therapy.
We have added the following to our manuscript:
“Abdominal computed tomography (CT) scan was performed which demonstrated extensive small bowel wall thickening with enhancement (target sign), and engorgement of visible mesenteric vessels (comb sign) with significant ascites; there were no ischaemic changes nor thrombosis of the mesenteric vessels.”
Round 2
Reviewer 1 Report
I do not see any significant improovement after revision of the article. Although it is rare this condition is well-known and does not represent any novelty. Unfortunately, I do not see any benefits for the readers of the Diagnostics from this report.
Author Response
We respect the Reviewer's opinion regarding the significance of our manuscript. While this condition might be well-known to senior and experienced clinicians such as our Reviewer, especially those in the field of rheumatology and internal medicine, the vague presenting symptoms of our patient nevertheless represents a diagnostic challenge to those less well-versed with SLE.
We would still like to assert the importance of our manuscript, which is to inform junior clinicians and specialists in other fields (e.g. emergency physicians, surgeons) of such a possibility. Recognising this possibility would hopefully help to avert unnecessary surgical intervention in similar cases presenting to the emergency department.
We sincerely hope that the Reviewer can reconsider our manuscript from such a perspective.
Reviewer 3 Report
Dear Authors,
Thank you for considering the comments and adjusting the manuscript.
Table 1 shows the laboratory findings at admission and after one month. Unfortunately, the table is missing the normal values in your laboratory.
The method for the determination of DNA antibodies is missing, probably ELISA (?).
You describe elevated DNA antibodies and ANA with "speckled" pattern.
DNA antibodies do not make a "speckled" pattern but a homogeneous pattern with positive mitoses "homogeneous" (AC-1)(anapatterns.org).
The speckled pattern can be divided into AC-2 "dense fine speckled", AC-3 "discrete specled/centromere", AC-4 "nuclear fine speckled", and AC-5 "nuclear large speckled". AC-2 (DFS70) is found in healthy individuals with high ANA titers without risk of ever developing CTD, AC-3 in scleroderma/CREST syndrome, AC-4 in Sjögren's syndrome, and AC-5 in overlap/sharp syndrome. Of course, patterns AC-3-AC-5 are also found in patients with SLE, but they should be differentiable as ENA. However, the absence of the "homogeneous" pattern (AC-1) on ANA IFT almost certainly excludes the presence of higher titer antibodies to DNA. This suggests that the result in the ELISA is false positive, which is possible in infections. This in turn makes the diagnosis of SLE questionable.
You added that you give hydroxychloroquine 200mg/day for 3 months in addition to steroids. If you believe in your diagnosis of SLE, you should give the HCQ permanently, as the main effect is not relapse treatment but prophylaxis.
Complement (C3/C4) is depleted during infections and there are people with congenital deficiency of C3 and/or C4 who cannot adequately synthesize more complement during infections.
In summary, the diagnosis of SLE and the consequence of the resulting therapy remains inconclusive for the expert and an infection with random immune phenomena remains not unlikely.
Author Response
We would like to thank the reviewer for their expert input in improving our manuscript. Our itemised responses are as below:
Question 1
Table 1 shows the laboratory findings at admission and after one month. Unfortunately, the table is missing the normal values in your laboratory.
Response 1
We have added our laboratory's reference ranges to Table 1 as recommended.
Question 2
The method for the determination of DNA antibodies is missing, probably ELISA (?).
Response 2
ELISA was indeed the method for our anti-dsDNA investigation. We have added that into our manuscript [line 82].
Question 3
You describe elevated DNA antibodies and ANA with "speckled" pattern.
DNA antibodies do not make a "speckled" pattern but a homogeneous pattern with positive mitoses "homogeneous" (AC-1)(anapatterns.org).
The speckled pattern can be divided into AC-2 "dense fine speckled", AC-3 "discrete specled/centromere", AC-4 "nuclear fine speckled", and AC-5 "nuclear large speckled". AC-2 (DFS70) is found in healthy individuals with high ANA titers without risk of ever developing CTD, AC-3 in scleroderma/CREST syndrome, AC-4 in Sjögren's syndrome, and AC-5 in overlap/sharp syndrome. Of course, patterns AC-3-AC-5 are also found in patients with SLE, but they should be differentiable as ENA. However, the absence of the "homogeneous" pattern (AC-1) on ANA IFT almost certainly excludes the presence of higher titer antibodies to DNA. This suggests that the result in the ELISA is false positive, which is possible in infections. This in turn makes the diagnosis of SLE questionable.
Response 3
We apologise for failing to specify the ANA pattern as AC-4,5. ENA testing for anti-Smith antibodies was elevated at 10 units/mL (reference range: <5 U/mL). We have added this information in Table 1.
Question 4
You added that you give hydroxychloroquine 200mg/day for 3 months in addition to steroids. If you believe in your diagnosis of SLE, you should give the HCQ permanently, as the main effect is not relapse treatment but prophylaxis.
Complement (C3/C4) is depleted during infections and there are people with congenital deficiency of C3 and/or C4 who cannot adequately synthesize more complement during infections.
In summary, the diagnosis of SLE and the consequence of the resulting therapy remains inconclusive for the expert and an infection with random immune phenomena remains not unlikely.
Response 4
We apologise for inadvertently creating the perception that our patient was treated with hydroxychloroquine (HCQ) for only 3 months. She was actually discharged with a 3-month supply of HCQ, with prescription refills given each clinic visit. We have corrected that statement to read simply "Oral hydroxychloroquine 200 mg once daily was also prescribed" [line 93] to avoid this confusion.
Although at 1 month follow-up the patient had complete resolution of gastrointestinal symptoms and minimal arthralgia, she subsequently developed flares of rashes and polyarthritis at 2 months post-discharge, which was satisfactorily treated with another course of steroids in addition to methotrexate. We are therefore confident of our diagnosis of SLE, with the likelihood of an infection with random immune phenomena being very small.
We have included this additional information as follows: “Our patient was followed up in the outpatient clinic a month post-discharge with complete resolution of gastrointestinal symptoms and minimal arthralgia, though at 2 months post-discharge she developed another flare of polyarthritis and rashes which required a second course of steroids with coadministration of methotrexate” [Lines 97 – 99].
We hope that these amendments are sufficient to convince our expert reviewer and readers that the diagnosis of SLE is accurate in our patient. Recognition of the possibility of SLE in young female patients presenting with abdominal pain and ascites to the emergency department is crucial in averting unnecessary surgical interventions.
Round 3
Reviewer 1 Report
I do not see any significant improovement after all revisions of the article. Although it is rare this condition is well-known and does not represent any novelty. Unfortunately, I do not see any benefits for the readers of the Diagnostics from this report.
Reviewer 3 Report
Thank you for the modification. The case report is in an acceptable form, even the discrepancy between the non homogeneous ANA pattern and DNS antibodies is not resolved.